# A 3D Tissue Model of Traumatic Brain Injury with Excitotoxicity That Is Inhibited by Chronic Exposure to Gabapentinoids

**DOI:** 10.3390/biom10081196

**Published:** 2020-08-17

**Authors:** Nicolas Rouleau, Mattia Bonzanni, Joshua D. Erndt-Marino, Katja Sievert, Camila G. Ramirez, William Rusk, Michael Levin, David L. Kaplan

**Affiliations:** 1Department of Biomedical Engineering, Science and Technology Center, 4 Colby Street, School of Engineering, Tufts University, Medford, MA 02155, USA; Nicolas.Rouleau@tufts.edu (N.R.); Mattia.Bonzanni@tufts.edu (M.B.); Joshua.Erndt_Marino@tufts.edu (J.D.E.-M.); Katja.Sievert@tufts.edu (K.S.); Camila.Ramirez@tufts.edu (C.G.R.); William.Rusk@tufts.edu (W.R.); 2Department of Biomedical Engineering, Initiative for Neural Science, Disease, and Engineering (INSciDE), Science & Engineering Complex, 200 College Avenue, Tufts University, Medford, MA 02155, USA; 3Department of Biology, Allen Discovery Center at Tufts University, Science & Engineering Complex, 200 College, Avenue, Medford, MA 021553, USA; michael.levin@tufts.edu; 4Wyss Institute for Biologically Inspired Engineering, Harvard University, Boston, MA 02115, USA

**Keywords:** traumatic brain injury, tissue engineering, excitotoxicity, 3D neural tissues, voltage-gated calcium channels

## Abstract

Injury progression associated with cerebral laceration is insidious. Following the initial trauma, brain tissues become hyperexcitable, begetting further damage that compounds the initial impact over time. Clinicians have adopted several strategies to mitigate the effects of secondary brain injury; however, higher throughput screening tools with modular flexibility are needed to expedite mechanistic studies and drug discovery that will contribute to the enhanced protection, repair, and even the regeneration of neural tissues. Here we present a novel bioengineered cortical brain model of traumatic brain injury (TBI) that displays characteristics of primary and secondary injury, including an outwardly radiating cell death phenotype and increased glutamate release with excitotoxic features. DNA content and tissue function were normalized by high-concentration, chronic administrations of gabapentinoids. Additional experiments suggested that the treatment effects were likely neuroprotective rather than regenerative, as evidenced by the drug-mediated decreases in cell excitability and an absence of drug-induced proliferation. We conclude that the present model of traumatic brain injury demonstrates validity and can serve as a customizable experimental platform to assess the individual contribution of cell types on TBI progression, as well as to screen anti-excitotoxic and pro-regenerative compounds.

## 1. Introduction

Traumatic brain injuries (TBIs) account for millions of hospitalizations every year and contribute to a third of all injury-related deaths in the United States [1,2,3]. Cerebral lacerations, injuries which distort or destroy brain matter, are among the most deleterious forms of TBI. Often the result of a penetrating head injury, cerebral lacerations are notoriously difficult to treat once incurred and often leave large intracerebral cavities in their wake that become fluid-filled and inhospitable to repair [4,5]. Soon after the initial injury is sustained, local chemical cascades are triggered at the wound site, generating secondary brain injuries [6,7,8]. The causes of secondary injury are various [9,10]; however, repeated and protracted membrane depolarizations are perhaps the most threatening to long-term brain function—A phenomenon that is known as excitotoxicity [11,12]. In particular, increased interstitial glutamate is thought to drive excitotoxic lesion formation in vivo [13,14]. Cycles of injury and re-injury caused by an insidious weakening of the brain’s capacity to tightly control neuronal excitability commonly devolves into epileptogenesis, causing additional lesions with each successive electrical paroxysm [15,16]. The net result of a severe brain injury is a prolonged state of tissue atrophy that exacerbates the brain’s normal rate of decay [17]. Though the typical individual will lose approximately 10% of their neocortical neurons over a lifetime, a TBI sufferer can expect to display signs and symptoms of accelerated aging, ultimately experiencing increased rates of dementia and other neurological consequences of neurodegeneration [18,19]. In fact, the brains of TBI sufferers were predicted to be 5 to 10 years older than expected, with evidence of chronic postinjury tissue loss [20]. Because primary injuries are impossible to anticipate and difficult to prevent, mitigating the damage associated with the secondary injury is central to any TBI treatment regimen and an essential first step toward prospects of brain injury reversal. However, valid models of TBI with which to conduct high-throughput screening and drug discovery are currently unavailable.

Acute TBI models in rodents have demonstrated the potent anti-excitotoxic effects of N-methyl-D-aspartate (NMDA) receptor antagonists, such as ketamine, on the course of brain damage following injury [21,22,23]. Spreading depolarization can be inhibited by ketamine in humans, suggesting a common mechanism across species [24]. Although ketamine does not promote regeneration at the site of injury, it does prevent secondary injury and post-injury seizure activity. The hypothesized mechanism involves a decrease in Ca^2+^ influx, preventing the sustained depolarization of neurons in the peri-injury region and inhibiting cell death despite increased interstitial glutamate. In principle, any treatment that prevents sustained depolarization or burst-like firing activity can prevent excitotoxicity, particularly those that target calcium influx. In fact, many authors have reported that voltage-gated calcium channel (VGCC) blockers—Including gabapentin (GBP) and pregabalin (PGB)—Have been used to treat secondary brain injury [25,26,27]. A valid model of TBI and its secondary injury characteristics should therefore be responsive to treatments that reduce excitability and limit calcium influx.

Most current efforts to study the many ways by which TBI-related degeneration can be prevented or mitigated are limited to low-throughput animal model systems, which are both costly and inefficient, as well as generally poorly representative of the human condition. Tissue engineering represents a compromise between in vivo and in vitro model systems, preserving complexity and translational relevance while maintaining flexibility, cost-effectiveness, and practicality [28]. Current in vivo culture models of TBI, including simulated injuries of transection, compression, hydrostatic pressure, fluid shear stress, shear strain, and stretch injuries in organotypic slice cultures, are highly predictive (88%) of observations in vivo with expressed hallmarks of secondary brain injuries [29]; however, their limited in vitro functions prevent insights into potential treatments. Indeed, organotypic slice cultures are essentially tissue preparations from living animals that are necessarily severely damaged by process of dissection and slicing [29]. Therefore, experimental designs involving organotypic slice cultures to study TBI cannot, by definition, include a true non-injured control condition.

We previously engineered a cortical brain model that recapitulated the major features of neural tissues incorporated rodent or human cells [30], and provided a suitable system to begin to study chronic neurodegenerative diseases [31,32]. TBIs were originally proposed as one potential application of the cortical brain model [28]. To explore applications for TBI research, cells derived from embryonic rat brains were seeded within porous, topologically complex silk protein-based scaffolds and later embedded in hydrogels that simulated the extracellular matrix (ECM) to provide a medium within which neural networks could form. The result was a highly interconnected, tunable, bioengineered cortical brain model, which could be adapted to study pathophysiology. Unlike most other model systems, our bioengineered neural tissues are sustainable in vitro for extended time frames and are highly modular, allowing experimenters to tailor every component of the tissue to accommodate the experimental protocol. The objective of the present study was to develop a laceration-based TBI model using our 3D bioengineered neural tissues that would express features of excitotoxicity and respond to current treatments of TBI in vivo, such as gabapentinoids.

## 2. Methods and Materials

### 2.1. 3D Neural Tissue Fabrication

Our original protocol delineates the fabrication process in great detail, outlining the scaffold preparation, cell seeding, hydrogel embedding, and maintenance procedures [33]. Similarly, reviews of silk fibroin biomaterial processing and extensive characterizations of the porous 3D scaffolds, including the compressive strength, can be found elsewhere [34,35,36]. We will, however, provide a brief summary of the main methods here. Rat embryos (Sprague–Dawley, Charles River) were harvested at E18 whereupon their cortices were resected to isolate neural and glial cells, as was previously characterized [37]. The Tufts University Institutional Animal Care and Use Committee (IACUC) approved all animal work according to NIH guidelines. Once isolated, 10^6^ cells (in media) were seeded in toroidal silk-fibroin scaffolds (silk concentration: 6% *w/v*, height: 2 mm, outer diameter: 6 mm, inner diameter: 2 mm) with 500–600 µm pores that were coated in poly-d-lysine (0.1 mg/mL). The seeding procedure was performed in 96-well plates, which were incubated (37 °C, 5% CO_2_, and 95% relative humidity) for 24 h. The cell-containing scaffolds were then transferred to new 96-well plates and embedded within a collagen type-I (rat tail) hydrogel (3 mg/mL). The structural integrity of the silk scaffold material and collagen-based hydrogel were recently confirmed to remain stable for over 2 years in culture conditions [38]. After a 30 min gelation period within the incubator, samples were transferred to 24-well plates and maintained in 1.5 mL of fresh medium composed of a Neurobasal™ medium (Gibco™, Thermo Fisher, Grand Island, NY, USA) with 1% *v/v*
l-glutamine (GlutaMAX™, Thermo Fisher, Waltham, MA, USA), 2% *v/v* B27 supplement (Thermo Fisher), and 1% *v/v* penicillin–streptomycin (Corning™, Cellgro™, Cambridge, MA, USA; antibiotic solution). Media were replaced (half of 1.5 mL) every 3 days. Once the neural tissues were assembled, neurons and glial cells formed complex network structures where the cell bodies remained primarily affixed to silk fibers, extending their neurites across the central window to form long-range (mm-scale) connections (Figure 1B,C).

### 2.2. Laceration Procedure

To simulate the laceration-based TBIs, we removed the central core of the neural tissue samples and filled the void with a cell-free, collagen-based (rat, type-I) hydrogel cap to detect the potential re-growth of neurites across the construct. High re-growth and repair without treatment would indicate poor model validity since TBIs cause lasting damage in vivo with limited regenerative potential. Biopsy punches (Acuderm Inc., Fort Lauderdale, FL, USA) with 2 mm and 3 mm blade diameters were used to core the samples. The 2 mm biopsy punch removed only the central mass of the hydrogel while the 3 mm biopsy punch removed both the hydrogel and inner lumen (toroid) of the silk scaffold containing large proportions of cell bodies (Figure 1D). Core removal was performed on polydimethylsiloxane (PDMS) platforms within biosafety cabinets under sterile conditions. Once lacerated, the samples were transferred to 96-well plates and re-embedded as described previously to cap their injury sites. After a 30 min incubation period to permit gelation, the injured samples were transferred to new wells and maintained as described previously. Non-injured controls were generated by placing samples on PDMS platforms for the same time period as the lacerate samples, transferring them to new 96-wells, and then returning them to media, thus accounting for changes due to temperature and manipulation. The extent to which the samples were damaged could then be assessed and quantified (Figure 1E,F).

### 2.3. Treatments

Once the injury was characterized, the lacerated samples were incubated for no longer than 10 days to assess the treatment conditions. Treatments were administered to determine whether the secondary features of the simulated injury could be mitigated pharmacologically. Gabapentin (GBP, Sigma-Aldrich, St. Louis, MO, USA) and pregabalin (PGB, Sigma-Aldrich, St. Louis, MO, USA) were selected as it was reasoned that by preventing excess depolarization and calcium uptake, re-injury could be prevented. GBP and PGB were selected because we sought to target the α2δ-1 subunit to block VGCCs—A mechanism shared by both compounds [39]—To mitigate putative excitotoxicity within the model system. The term “gabapentinoid” (i.e., GBP and PGB) and “VGCC blockers” are used interchangeably throughout the text. We incorporated the use of the NMDAR antagonist AP5 (Sigma-Aldrich, St Louis, MO, USA) as a rescue drug in several experiments, as we hypothesized it would serve as a positive control to mitigate excitotoxicity by reducing post-injury Ca^+2^ influx.

Two administration schedules were used during the post-TBI treatment phase. The first schedule was an acute exposure beginning immediately after the injury and a re-embedding that lasted 24 h, followed by a withdrawal of the treatment and a return to regular media. The second schedule consisted of a chronic exposure that was first administered immediately after the injury and re-embedding, with re-administrations of the drug with fresh media every 3 days for 10 days post-injury.

### 2.4. DNA Quantification

To quantify post-injury cell death within the 3D samples, DNA was extracted and quantified. A Quant-iT-PicoGreen dsDNA Assay kit (Thermo Fisher, Waltham, MA, USA) was used to evaluate the DNA concentration (ng/mL) according to the manufacturer’s protocol. To briefly summarize the protocol, samples were placed in conical tubes containing 200 μL of a 0.05% Triton-X 100 solution (in dH_2_O) to permeabilize the cells and extract the DNA content. Once immersed, the samples were frozen and maintained at (−80 °C) until assayed. After thawing the samples, they were shredded, centrifuged, and 100 μL of the supernatant was transferred to a 96-well, black-walled plate. Standards (0–2000 ng/mL) were prepared in duplicate and added to the same plate. Wells were then filled with an equal volume of a 1:200 dilution of PicoGreen^®^ reagent (Thermo Fisher, Waltham, MA, USA). The plate was then inserted into a SpectraMax M3 Microplate Reader (Molecular Devices, San Jose, CA, USA) and the fluorescent intensity was measured (ex/em: 485/538) using SoftMax Pro (Molecular Devices, San Jose, CA, USA) (v.7.0.2). Data were extracted, background-corrected, standardized, and transferred to IBM SPSSv20 Pro (Armonk, NY, USA, for analysis).

### 2.5. Glutamate Quantification

A colorimetric Glutamate Assay kit (Abcam, Cambridge, MA, USA) was utilized to perform the quantification. Our pilot data indicated that the standards prepared in media and the manufacturer’s assay buffer were associated with highly correlated absorbance values (*r* = 0.99, *p* < 0.001; data not shown); moving forward, the media served as both the buffer and background condition. Standards were generated (0–10 nmol/well) and added (50 μL) to the flat bottom standard 96-well plates in duplicate. Media samples were added to the same plates. A Reaction Mix composed of a Glutamate Enzyme Mix and Glutamate Developer was added to each well, mixed, and incubated at 37 °C in the dark for 30 min. Absorbance (450 nm) was measured using a SpectraMax M3 Microplate Reader with SoftMax Pro (v.7.0.2). Data were extracted, corrected for background, standardized to a time-matched average DNA concentration for independent groups, and transferred to IBM SPSSv20 for analysis.

### 2.6. Immunocytochemistry

Constructs were transferred from the media to Dulbecco’s phosphate-buffered saline (DPBS) (Sigma-Aldrich, St. Louis, MO, USA), washed three times, and placed in 4% paraformaldehyde (PFA) (Santa Cruz, Dallas, TX, USA) overnight to allow full perfusion and fixation. Once fixed, samples were washed in DPBS and transferred to a blocking solution (2% *v/v* donkey serum, 0.2% *v/v* Triton X-100, 20 mg/mL bovine serum albumin in DPBS) and incubated at room temperature for 1 h. Samples were then immersed in a fresh blocking solution that contained the diluted primary antibodies (1:1000) for 24 h at 4 °C (mouse anti-βIII Tubulin antibody (TUJ1, Thermo Fisher Scientific, Waltham, MA, USA). Secondary antibodies (Alex 488 anti-mouse) (Thermo Fisher Scientific, Waltham, MA, USA) were then diluted (1:250) and applied for 24 h at room temperature. Finally, samples were washed (in the dark) in DPBS, stained with DAPI (300 µM) for 30 min, washed again, and then maintained in DPBS at 4 °C in the dark until they were imaged.

A Leica TCS SP8 Confocal Microscope (Leica, Wetzlar, Germany) equipped with photomultiplier tubes (400–800 nm) was used to perform the fluorescence imaging. Each sample was positioned onto a glass-bottom 35 mm dish (Corning) and covered with a small amount of DPBS. We elected to image samples under water immersion and at high magnification. Each composite image that was acquired consisted of a z-stack (200 µm) with 4 µm steps. As reported previously [40], the silk scaffold displays auto-fluorescence under short wavelengths and was therefore present when simultaneously imaging the DAPI-tagged cells. The z-stack, maximum-projection images were manually loaded to Image J (Public Domain, BSD-2) (version 1.47), converted to 8-bit binary images, despeckled, and inverted. Area was then measured.

### 2.7. Live Cell Imaging

To visualize cell viability and patterns of cell loss over time as a function of the injury, samples were stained with Calcein AM (Thermo Fisher Scientific, Waltham, MA, USA) (ex/em: 495/515), a cell-permeant dye that fluoresces intracellularly—Indicating the presence of live cells. A Nikon Eclipse Ti2 Inverted Microscope System with a Zyla 5.5 sCMOS (Andor, Belfast, UK) was used to image the live cells within the tissue constructs. To measure the irregular surface area of the scaffolds, 100 µm z-stacks (1 µm slices) were assembled into maximum-projection representations. Images were focused and the backgrounds were subtracted in NIS-Elements (Nikon, Melville, NY, USA). Monochromatic data were uniformly assigned RGB values in ImageJ (version 1.47) to colorize the images.

### 2.8. Local Field Potentials

Tissue function was assessed by evaluating the spontaneous electrical activity within the samples before and after the experimental injury. Our original electrophysiological protocol is provided in detail elsewhere [41]; however, we will re-iterate the essential methods here. Each 3D sample was transferred to a 35 mm plastic petri dish (Corning) that contained 2 mL of an extracellular solution composed of the following elements (mM): 130 NaCl, 1.25 NaH_2_PO_4_, 1.8 MgSO_4_, 1.6 CaCl_2_, 3 KCl, 10 HEPES-NaOH, 5.5 glucose, pH 7.4. Samples were placed on a WP-16 Warmed Platform (Warner Instruments, Hamden, CT, USA) controlled by a TC-134A Handheld Temperature Controller and set to maintain an extracellular solution temperature of 37 °C.

With a reference electrode inserted into the extracellular solution at the periphery of the dish, samples were immobilized using a stabilizing rod. The recording electrode consisted of a filament inserted into a borosilicate glass pipette (40–80 MΩ), which was pulled using a Sutter P-97 (Novato, CA, USA). A head stage was used to position the probe within 50 µm of silk fibers within the main body of the scaffold, equidistant between the inner lumen of the central window and the outer edge of the sample. Once in position, the LFPs (mV) were collected at 2500 Hz for 5 min under baseline conditions to observe spontaneous electrical activity. An Axon Instruments analog-to-digital converted the received signals from an Intan digital amplifier. The recorded traces, which were represented in Clampex 10.7 (Axon Instruments, Molecular Devices, San Jose, CA, USA), were exported to Clampfit 10.7 to perform threshold-based spike potential isolation. Spikes were defined as fluctuations exceeding ±0.35 mV, which was equivalent to an extreme z-score between 3 and 9. Spike potential frequencies were computed (LFPs/min) and served as the final measure of activity.

### 2.9. Excitotoxicity Assay

To confirm the neuroprotective effects of the VGCC blockers on our model of TBI, we performed an excitotoxicity assay on the primary cortical monolayers (2D) with a narrow subset of potential treatment conditions using the same cells that formed our 3D neural tissues. Cortical cells isolated from E18 embryonic rats were seeded in 24-well plates coated in PDL (3.3 × 10^5^ cells/well). Cells were maintained for 2 weeks to promote a dense, mature network with fresh media supplied every 3–4 days. After the maturation period, cells were exposed to either 100 µM glutamate, 100 µM NMDA, or an equivalent volume of vehicle (dH_2_O) in media. Glutamate was prepared by solubilizing powdered l-glutamic acid ≥ 99% (HPLC, Thermo Fisher Scientific, Waltham, MA, USA) dissolved in water with constant stirring and applied heat. Having previously identified high concentrations (100 µM) of GBP and PGB as preventative of cell death in our 3D TBI model, we hypothesized that glutamatergic excitotoxicity could be prevented by the same drugs.

### 2.10. Patch-Clamp Electrophysiology

Patch clamp experiments in the whole cell configuration were carried out after 14 days of culture at 37 °C ± 1 °C. The neurons were superfused with an extracellular-like solution containing (in mM) 130 NaCl, 3 KCl, 1.6 CaCl_2,_ 1.8 MgCl_2_, 1.25 NaH_2_PO_4_, 10 HEPES NaOH, 5.5 Glucose; pH = 7.4. The pipettes had a resistance of 5–9 MΩ and were filled with an intracellular-like solution containing (in mM) 130 K-Asp, 10 NaCl, 5 EGTA-KOH, 2 MgCl_2_, 2 CaCl_2_, 2 ATP (Na-salt), 5 creatine phosphate, 0.1 GTP, 10 HEPES-KOH; pH 7.2. Similar to the 3D samples that were exposed to a 24 h immersion within media supplemented with either 100 µM of GBP or PGB, the 2D monolayers received a 24 h exposure whereupon they were returned to normal media and measured 7 days later. In a separate condition, the cells were exposed to the same concentrations of the drugs while being recorded to characterize the acute biophysical effects. In the chronic condition, the drugs were maintained in the cell media for 7 days before recording. To investigate neuronal excitability, 2500 ms steps of injected current in the –20/+80 pA range were applied in the current clamp mode from a holding voltage of –70 mV. The spike threshold was defined as the first current step able to induce action potential firing. The resting membrane potential (RMP) was recorded in the I/O configuration. Neither series resistance compensation nor leak or liquid junctional potential corrections were applied. All protocols were designed using pClamp 10.2 (Axon) and the data were analyzed using Origin Pro 9 (Origin Lab, Miami, FL, USA).

### 2.11. Cell Cycle Analysis with Flow Cytometry and Propidium Iodide

Cell cycle assessment was performed 24 h post injury to achieve a snapshot of the cell state after their respective treatments. Analysis was performed utilizing the propidium iodide (PI) flow cytometry kit (Abcam, Cambridge, MA, USA), essentially following the manufacturer’s protocol. Briefly, cells were lifted with TrypLE Express (Gibco, Grand Island, NY, USA) treatment for 5–10 min at 37 °C. The duration of the cell lifting varied in the experiment but was monitored by eye periodically for cell sheet lifting. Lifted cells were pelleted with 500 g for 5 min and subsequently fixed by resuspension in 400 µL of ice-cold PBS followed by addition of 800 µL of ice-cold 100% EtOH. On the day of analysis, the fixed cells were pelleted at 500 g for 5 min followed by resuspension in 300 µL of a 1× PI solution with 1× RNase at 37 °C for 30 min. The cells labeled with PI were then analyzed using a flow cytometer (BD FACS Calibur, Franklin Lakes, NJ, USA) with a 488 nm excitation and the FL-2 emission filter. Cell-cycle gates were selected manually on one representative trace for the experiment and then the same gates were applied across samples to calculate the percentages of cells in certain stages of the cell cycle. Appendix A shows the gating strategy, which involved gating fixed cells by size and granularity, excluding debris, and defining the cell cycle phases for G1, S, and G2-M.

## 3. Results

### 3.1. A 3D Model of TBI that Expresses an Excitotoxic Phenotype

We used our 3D bioengineered cortical brain model seeded with embryonic rat neurons and glia as a platform to study TBI (Figure 1A,B). As a metric of injury severity, neurite re-growth within the hydrogel region was quantified as a function of the diameter of the laceration. A TUJ1-positive signal projected across the z-stacks of fixed width was quantified to obtain the neurite area. Normal infiltration of neurites into the original hydrogel after cell seeding can be observed in Figure 1. A one-way ANOVA revealed an effect of time on neurite area, *p* < 0.05 (Figure 1C). The pattern of neurite extension and growth reflected an initial increase over 7 days followed by a plateau phase punctuated by increased variability between the samples. Based upon this observation and to ensure complete neurite infiltration among all the samples in subsequent experiments, lacerations were delivered 14 days post-seeding.

Once lacerated, whether by a 2 mm or 3 mm biopsy punch, the neurite signal was completely eliminated from the central, re-filled hydrogel window (CW). After 3 days, however, differences between the laceration groups became evident as a function of time, *p* < 0.001, *η*^2^ = 0.72 (Figure 1E,F). The samples that were injured using a 2 mm biopsy punches displayed significant re-growth at 3 days, 7 days, and 14 days post-injury. Whether samples were lacerated with a 2 mm biopsy punch 3, 7, or 14 days post-seeding, re-growth of the neurites within the central window was evident within 3 days post-injury, returning to baseline levels and plateauing thereafter (Appendix A). However, when lacerated with a 3 mm biopsy punch, which in addition to the hydrogel also damaged the scaffold region where most cell bodies were located, samples were unlikely to display any neurite re-growth within the CW. Indeed, there was no significant difference between neurite area as inferred by TUJ1-positive staining within the hydrogel region on the day of injury and 14 days later (*p >* 0.05). These results suggest that the 2 mm condition was a relatively mild injury as compared to the more severe 3 mm injury. To more accurately reflect the in vivo phenotype, we continued our experiments using the 3 mm injury condition that was not associated with neurite re-growth.

To further characterize the severity of the 3 mm injury condition, the DNA concentration was quantified at fixed intervals following injury and re-embedding: 0 day (immediately, *n =* 26), 3 days (*n =* 24), 7 days (*n =* 24), and 10 days (*n =* 28). A two-way ANOVA revealed the main effects of injury (*p* < 0.001) and time (*p* < 0.001) on DNA concentration, explaining 34% and 12% of the variance, respectively. The primary injury was associated with an initial (0 day) 64% reduction in DNA concentration relative to the controls, which was followed by an additional 17% over the proceeding 10 days (81% total) (Figure 2A). For comparison, the non-injured, control samples incurred a 55% reduction in DNA concentration over 10 days, representing the normal deterioration of the system. These data suggest that the severe injury was associated with increased cell dropouts as inferred by significant DNA reductions after the initial injury, which were followed by more gradual decrements over time. Live imaging revealed the pattern of cell dropout was spatially predictable, where regions most proximal to the injured core became more cell-sparse over time (Figure 2B).

We predicted that the glutamate concentrations in the media would increase following the primary injury, recapitulating the excitotoxic phenotype typically observed in vivo. After injury and re-embedding, samples were returned to fresh media and incubated for 4 days before the media was replaced as part of the regular maintenance procedure. Glutamate concentrations in media (normalized to DNA) associated with injured and non-injured samples were recorded at 24 h (*n =* 18) and 3 d post-injury (*n* = 18) and compared. In addition, we collected media once again 7 d post-injury (*n* = 18) after completely replacing the media on Day 4, allowing 3 days of incubation and glutamate accumulation. A two-way ANOVA revealed the main effects of injury (*p <* 0.001) and time (*p* < 0.001) upon glutamate concentrations, explaining 44% and 24% of the variance, respectively (Figure 2C). Glutamate concentrations at 24 h increased by 210% in media extracted from wells containing the injured samples relative to the fresh media, *p* < 0.001, *r*^2^ = 0.88. The control samples, by contrast, did not express significantly increased glutamate concentrations in their media relative to the fresh media 24 h later, *p* > 0.05. The injured samples expressed an increased glutamate concentration in the media at 3 d relative to the non-injured controls, *p* < 0.001, *r*^2^ = 0.70. The effect persisted to a lesser degree at 7 d after replacing the wells containing tissue samples with fresh media (indicated by the gray line, Figure 2C) and incubating them for 3 additional days, *p* < 0.005, *r*^2^ = 0.43. These data indicate the injured samples expressed higher glutamate concentrations relative to the non-injured controls—a trend that persisted even after exchanging the consumed media with fresh media, suggesting an enduring effect of TBI on the glutamate concentrations in the media.

Before sustaining the injury, the samples (*n* = 5) contained live cells distributed throughout both the scaffold and hydrogel region—both the top and edge surfaces of the constructs expressed distributed CA-positive staining (Figure 2D). At the 24 h time point following the laceration, the peri-injury region did not contain live cells. Over time, the peri-injury region occupied greater proportions of the central mass of the tissue samples. It was evident that the edge surface, the most distal region of the scaffold relative to the injured cored, was not affected by the injury as inferred by live imaging. After 7 d, the pattern of live cell loss progressed such that only small patches of cells were observed with a receding front extending to the outer edges of the top surface. These qualitative data indicated that there was a predictable spatial pattern of cell dropout following injury; that is, cell death propagated outward from the core, only minimally affecting the most distal periphery.

Next, our goal was to mitigate the damage and potentially promote the re-growth within the tissue samples by adding targeted compounds in the media at varying concentrations and for different durations post-injury. These experiments would test the hypothesized excitotoxic model while providing construct validity by using classes of compounds that are known to prevent brain damage in vivo.

### 3.2. TBI-Induced Cell Death Was Mitigated by Chronic Exposure to Gabapentinoids

Because GBP is known to act upon other targets, including HCN channels [42,43], we electrophysiologically characterized the immediate and sustained effects of both drugs (high concentration GBP and PGB, 100 µM) with monolayers (2D) of the same primary cortical neurons described previously to verify their net impact on neuronal excitability (Appendix A). Once it was clear that excitability could be affected, 3D tissues were exposed where the injured samples were incubated in their respective treatment conditions for up to 10 days post-injury before the terminal assessments were conducted (Figure 3A). A one-way ANOVA revealed an effect of treatment on DNA concentration, *p* = 0.001, *η*^2^ = 0.28 (Figure 3B). The major sources of variance were differences between the untreated injured samples and chronic (10d), high-concentration (100 µM) PGB (*p =* 0.006, *r*^2^ = 0.38) and GBP (*p* < 0.05, *r*^2^ = 0.22) conditions (*n* = 9/group). Live imaging confirmed the presence of increased cell distributions throughout the scaffolds of the treated samples (Figure 3D). Samples that were acutely (immediate 24 h exposure) treated with GBP or PGB at any concentration did not differ from the injured samples that were untreated (*p >* 0.05). There were no differences in DNA concentration between the non-injured samples and those that were injured and subsequently treated chronically (10 days) with either 100 µM GBP (*p >* 0.05) or 100 µM PGB (*p >* 0.05), suggesting the treatments normalized the DNA concentration. Though the injured samples treated chronically with 100 µM AP5 displayed similar concentrations of DNA relative to the non-injured controls, as well as to those that were injured and treated, the group variability was high and no significant increases in DNA relative to the injured samples could be discerned (*p* > 0.05). Immunocytochemistry revealed minor re-infiltration of neurites into the collagen window following injury and subsequent treatment conditions; however, debris fields obscured imaging within the scaffold region.

To account for drug toxicity, some samples were exposed to the treatment conditions without incurring a laceration injury (*n* = 9/group). The DNA concentration within samples was quantified 10 days after initiating the treatment, revealing an effect of treatment on cell dropout not due to injury, *p* = 0.009, *η*^2^ = 0.31 (Figure 3C). Chronic treatment of 100 µM AP5 (*p* = 0.04) and, perhaps, 100 µM PGB (*p* = 0.06, marginal effect), inhibited the cell dropout over the 10-day period. Together, the analyses of DNA concentration as a function of injury and treatment conditions revealed that chronic exposure to higher concentrations of VGCC blockers and NMDA receptor antagonists can mitigate cell dropout in our 3D tissue models. Indeed, injury-mediated cell dropout was most effectively inhibited by the VGCC blockers whereas the physiological decreases not associated with injury were inhibited by the NMDA receptor antagonists.

### 3.3. Spontaneous Electrical Activity Was Restored in TBI Samples Treated with Gabapentinoids

Functional assessments of the injured samples were performed immediately after inducing the injury as well as 10 days post-injury with or without treatment. Spontaneous LFPs were recorded (*n* = 6/group) over a 5 min period to determine the relative level of functional activity within the sample groups. A *t*-test revealed a marginal effect of injury immediately following the laceration procedure (*p =* 0.06). The functional decrement became evident at 10 d post-injury, *p* = 0.03, *r*^2^ = 0.38. A one-way ANOVA revealed an effect of treatment among the injured samples, *p* < 0.05, *η*^2^ = 0.30 (Figure 3E). Homogeneous subsets indicated that the major source of variance was the difference in spontaneous LFP expression within the non-treated injured samples and those that were treated chronically (10 day exposure) with 100 µM PGB (Figure 3F); however, all treatment conditions displayed increased LFPs/min relative to the injured samples that were not treated, with the exception of the chronic, 50 µM PGB condition. Together, the data indicate that tissue function was impaired by the induced injury and that spontaneous electrical activity could be restored when treated with VGCC blockers at most concentrations.

### 3.4. Neuroprotection as a Reduction of Neuronal Excitability

Results associated with 3D experiments suggested that more cells were present in injured tissue constructs when they were chronically treated with VGCC blockers. AP5 was similarly effective at increasing the DNA concentrations in samples that were not injured. On the basis of these results, we hypothesized two underlying mechanisms, which were not mutually exclusive. Increased live cell populations could be explained by enhanced neuroprotection, proliferation, or a combination thereof. To address the neuroprotective capacity of the chronic exposure conditions, we added excitotoxic compounds, glutamate or NMDA, to the wells containing neurons plated in 2D, which were simultaneously exposed to treatment conditions.

The untreated controls (*n* = 10) displayed decreased live–dead cell ratios when exposed to 100 µM glutamate (*n* = 10) and 100 µM NDMA (*n* = 10) relative to the wells containing media only, *p* < 0.001, *η*^2^ = 0.61 (Appendix A). This effect unsurprisingly confirmed that both glutamate and NMDA were indeed excitotoxic, contributing to increased cell death. Next, a two-way ANOVA revealed a significant interaction between the excitotoxic compounds and treatments on the live–dead cell ratios, *p* < 0.001, *η*^2^ = 0.23 (Figure 4). One-way ANOVAs and post-hoc tests were then computed for selected cases to isolate the contribution of each factor to cell viability.

When selecting for cases that were not associated with an excitotoxic compound (*n* = 40), a one-way ANOVA revealed a significant effect of treatment, *p* < 0.001, *η*^2^ = 0.62 (Figure 4A, left). Post-hoc tests revealed the major source of variance was a difference between AP5 and all other conditions (*p <* 0.001). When selecting for cases associated with an exposure to NMDA (*n* = 40), a similar effect was observed, *p* < 0.001, *η*^2^ = 0.51 (Figure 4A, middle). Again, a difference between AP5 and all other conditions was the major source of variance; however, increased cell viability was also noted for NMDA-exposed samples that were treated with PGB (100 µM, chronic) relative to vehicle (control), *p <* 0.05. GBP did not prevent NMDA-induced toxicity (*p >* 0.05). Finally, selecting for cases associated with an exposure to glutamate (*n* = 40), a unique effect of treatment was observed, *p* < 0.001, *η*^2^ = 0.44 (Figure 4A, right). Post-hoc tests revealed three homogeneous subsets: 1) AP5 and vehicle control; 2) GBP; and 3) PGB. PGB provided the greatest resistance to glutamate-induced toxicity (*p* < 0.001, *r*^2^ = 0.55), followed by GBP (*p* = 0.01, *r*^2^ = 0.31). AP5 did not prevent glutamate-induced toxicity (*p* > 0.05). Together, these data indicate that the different excitotoxic injuries were selectively inhibited by NMDAR antagonists and VGCC blockers. Only PGB was effective at preventing excitotoxicity associated with both NMDA and glutamate.

### 3.5. Primary Neuronal Monolayers Did Not Display Signs of Increased Proliferation after Treatment

Beyond neuroprotection, we hypothesized that increased DNA retention with 3D scaffolds could be due to enhanced proliferation associated with the simulated injury, the treatments, or an interaction between these conditions. These experiments were performed utilizing a 2D equivalent of our 3D laceration model because many quantitative cell cycle and proliferation analyses are difficult to perform in 3D. Using a P200 micropipette tip, we scratched selected wells with an “X” in the center of each well to deliver a simulated injury in 2D. Wells with and without scratches were also administered high concentrations (100 µM) of GBP (100 µM) or PGB (100 µM) and AP5 (100 µM). As expected from rat E18 dissociated cortical cell cultures [44], the cell cycle analysis revealed that a significant majority (~75%) of the cells were in a non-proliferative G1/G0 state. In addition, a two-way ANOVA revealed there was no interaction between the scratch and drug conditions. Similarly, there was no main effect of the scratch condition in any of the cell cycle phases (Figure 4B). In contrast, AP5 treatment, irrespective of the presence or absence of a scratch (no significant interaction was present), resulted in significantly enhanced levels of cells in G1/G0 (~80%, *p <* 0.0005) relative to all other experimental groups (~68%; Figure 4B). This was accompanied by significantly fewer cells in both the S-phase (~8%; *p <* 0.0005) and G2/M-phase (~12%, *p <* 0.0005), the cell cycle stages associated with proliferation (Figure 4B). Cumulatively, these data suggest 1) that the gabapentinoids do not have any clear proliferative effects, at least in 2D; and 2) that AP5 is acting distinctly from the other treatment modalities.

### 3.6. Chronic PGB Exposure Induced Sustained Reductions in Neuronal Excitability

We hypothesized that the VGCCs were reducing cell excitability, thus providing cells with anti-excitotoxic neuroprotection. To verify the effect of the chronic exposures to high concentrations of VGCCs blockers, we performed patch clamp experiments in the whole cell configuration (see Materials and Method). As seen in Figure 5A, representative action potential records show that a +50 pA current injection (holding potential at −70 mV) was sufficient, for all conditions, to elicit firing in the neurons. However, the firing pattern is different, as showed in Figure 5B, in which the GBP condition showed a reduced number of action potentials, but not in the shift of the curve. On the other hand, the PGB condition showed both a reduction of the firing capability as well as a rightward shift of the curve. The shift is in line with the significant increase of the spike threshold in the PGB condition (Figure 5C); the resting membrane potential was, instead, unaffected by the treatments (Figure 5D). Overall, the results indicated a decreased excitability due to the chronic treatments.

## 4. Discussion

One of the primary challenges to overcome when designing any model of pathology is to satisfy the fundamental criteria underlying the phenomenon in vivo. Of course, any reductionist model will necessarily neglect certain features of the modelled system; however, the essential features should be preserved to maintain the potential for translation. With TBI, it is essential to demonstrate that the cell or tissue model displays an enduring injury phenotype with expected biochemical and functional correlates. Further, the TBI model should be sensitive to treatments that are known to mitigate signs and symptoms of the pathology in vivo. The model we have described here satisfies the aforementioned criteria, displaying features of both primary and secondary injury that are mitigated by neuroprotective, anti-excitotoxic VGCCs. We specifically focused on the role of excitotoxicity, which is a well-documented phenomenon in animal models [45,46,47] and human cases [13,48].

Of particular relevance to TBI sequelae in vivo, increased glutamate release coupled with time-dependent cell dropout as inferred by decreased DNA concentration reinforced the validity of the model system (Figure 2). Even after replacing the media within which both injured and non-injured samples were immersed, the injury-related glutamate increases were sustained, indicating an effect on cells beyond the initial insult—A hallmark of secondary brain injury (Figure 2C). Of note, the glutamate concentrations associated with the control group (non-injured) inflected between 3 and 7 days after the reference day of injury. Likewise, we noticed decrements in DNA associated with the controls between 7 and 10 days from the same reference day (14 days post-seeding) (Figure 2A). We interpret these findings to reflect one limitation associated with the long-term culture of primary cortical cells derived from rodents. After approximately 21–24 days post-seeding, the 3D constructs began to slowly deteriorate whether injured or not. Interestingly, chronic administrations of 100 µM AP5 in both 3D (Figure 3C) and 2D (Figure 4A) prevented these expected, physiological decrements that were not associated with a deliberate injury—A result that recapitulates the expected anti-excitotoxic therapies in animal models [49,50]. Rodent-based neural tissue systems are therefore only suitable for short-term (<21 day) experiments. Human-induced pluripotent stem cells (iPSC) and neural stem cells (HNSC) are more suitable for long-term applications and can be readily integrated within the 3D silk scaffolds [30,31].

We have hypothesized that decreased excitability due to gabapentinoid exposure was effectively neuroprotective and not proliferative (Figure 4B), ultimately maintaining more live cell populations in each scaffold relative to the untreated controls. The excitotoxicity assay (Figure 4A) confirmed that, unlike AP5 and GBP, PGB could blunt the deleterious effects of the NMDA- and glutamate-mediated toxicity. AP5, though a potent antagonist of NMDA receptors, does not universally inhibit glutamate from binding other types of receptors. Likewise, GBP is simultaneously a VGCC blocker and a ligand at several other types of likely receptor targets, including HCN channels [42] as well as NMDA, AMPA, adenosine, and muscarinic receptors [51]. Together, these results suggest the VGCC blocker PGB was neuroprotective in our model of TBI, validating one of potentially many mechanisms underlying the secondary injury phase.

Pregabalin, which is selective for the α2δ-1 subunit of the VGCCs [39] and is known to decrease calcium-based activity [52], has also been shown to increase glutamic acid decarboxylase (GAD) at higher concentrations [3]. GAD is an enzyme that drives the decarboxylation of glutamate into γ-aminobutyric acid (GABA), increasing the availability of this inhibitory neurotransmitter while reducing the concentration of glutamate and ultimately dampening neuronal excitability. Similar effects upon GAD have been reported with administrations of gabapentin [53]. As glutamate-induced neuronal excitotoxicity is well-known to induce both necrosis and apoptosis [54,55,56], one potential mechanism underlying the observed neuroprotective effect is the GAD-mediated conversion of glutamate into GABA. Because necroptosis is sensitive to calcium influx, repeated and protracted depolarization events parallel the death pathways and contribute to complex TBI-induced secondary injury sequelae [54]. In consideration of these well-known mechanisms, the experiments presented here demonstrate that a minimal TBI model can be achieved and that compounds that prevent calcium influx can be applied to mitigate expected secondary injury processes.

The use of embryonic rodent cells to model TBI with relevance to the adult brain is worth some consideration. Because adult rodent neural cell cultures are generally inferior (i.e., poor longevity and cell integrity) to both embryonic and perinatal primary cell sources, we did not adapt the model system to include adult tissues. Consequently, the results should be interpreted with caution since the cells were embryonic and thus intrinsically more plastic. However, it should be noted that the cells did display robust electrophysiological activity (Appendix A) and the presence of voltage-gated channels can be inferred from the data presented in Figure 3. Based upon the data presented in Figure 4B, the primary cultures were also post-mitotic, which is characteristic of the adult phenotype.

Beyond limitations of longevity associated with the use of rodent cells, there are some fundamental parameters of the model system that should be considered. The space-occupying presence of the silk scaffold itself, lower cell densities than those observed in natural neural tissues, and simplified ECM composition (i.e., collagen only versus a heterogenous substrate) are intrinsic limitations that distinguish the current system from animal models. However, the flexibility of our bioengineered platform is such that multivariate mechanisms of TBI could be investigated by systematically incorporating glial cells, vasculature, and other brain-relevant factors (e.g., extracellular matrix, microbiome, hormonal states, diseased cells) into the design of the tissue constructs. This approach would represent a marked advantage over organoids, which are comparatively undesirable by dint of their rigid developmental phenotypes (i.e., normal but fixed morphogenic blueprints) that are not subject to selective insertion or deletion of neural tissue elements. Similarly, the current silk scaffold-based system represents a significant advance compared to simple hydrogel models that have previously been applied to study TBI [57]. Indeed, our scaffold–hydrogel hybrid structure allows the user to control the material properties, including the porosity and stiffness, to generate internal compartments, and to deliver significant lacerations while preserving the contour of the simulated tissues. Variations of the current platform will facilitate the isolation of the unique causative contribution of those factors that have been suspected to play an important role in secondary brain injury progression (e.g., reactive gliosis, vascular dysfunction). Studies such as these are equivalent to knock-out experiments whereby a discrete element, usually a gene, but a tissue component in this case, can be selectively removed or added to the construct to evaluate its individual contribution to the multivariate nature of the TBI. Interactions between elements would also be subject to investigation, inspiring new combinatory therapies to treat brain damage with multiple mechanisms of action. Another unique advantage of this platform is that it will enable the development of closed-loop control systems, which use microfluidic and optogenetic regimes based on real-time readouts of tissue physiology to provide on-the-fly stimulation and adjustment of interventions to manage the course of the healing response.

## Figures and Tables

**Figure 1 biomolecules-10-01196-f001:**
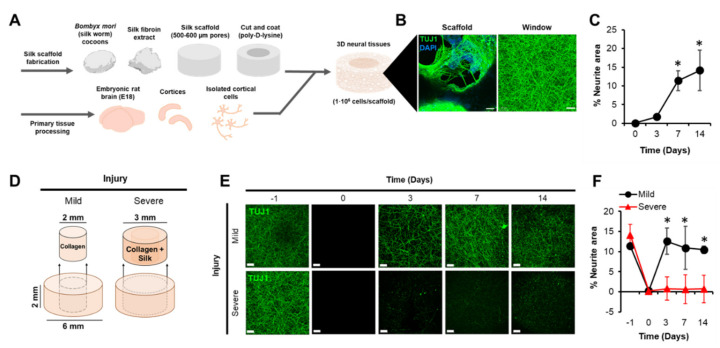
3D neural tissues were fabricated and injured to model a laceration-based traumatic brain injury (TBI). (**A**) Primary cortical neurons were harvested from dissociated embryonic (E18) rat cortices and injected into a toroidal, silk-based scaffold. (**B**) Both the Scaffold and Window regions contained a diffuse network of neurites as revealed by immunocytochemistry (TUJ1 green, DAPI blue). (**C**) Quantifying neurites within the Window region (hydrogel), it was evident that the maximum area occupancy was achieved after approximately 7 days of growth (* *p* < 0.05). (**D**) Mild and severe injuries were delivered with 2 mm or 3 mm surgical biopsy punches to the 3D neural tissue samples 14 days after seeding. (**E**,**F**) Neurite re-growth within the central window was evident after 3 days for mildly injured samples (* *p* < 0.05) whereas severely injured samples displayed a more permanent damage phenotype. All image scale bars represent 50 µm.

**Figure 2 biomolecules-10-01196-f002:**
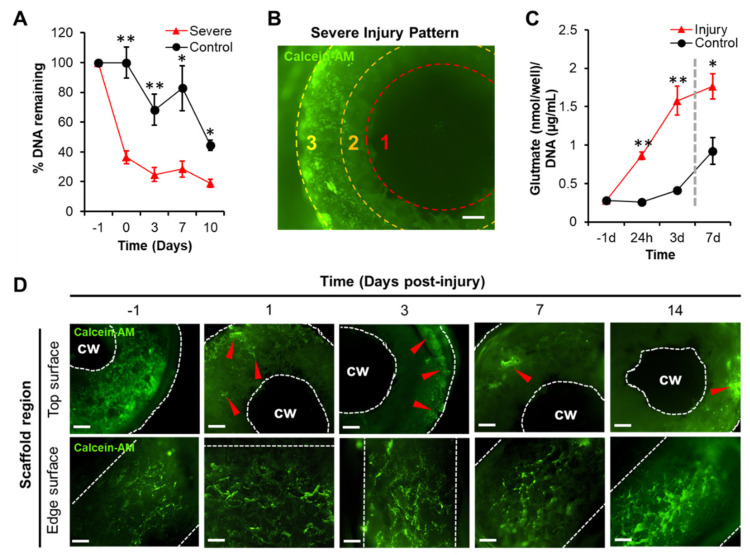
TBI model characterization to determine the injury severity, time-course, and biochemical changes. (**A**) DNA concentration within samples decreased significantly over time for the injured samples relative to the non-injured controls (*n =* 102 samples total); Day 0 (** *p* < 0.001), Day 3 (** *p* < 0.001), Day 7 (* *p* < 0.005), and Day 10 (* *p* < 0.005). (**B**) Live imaging revealed a center-surround pattern of cell death wherein the initial injury physically removed cells from the central-most portion of the scaffold (red, inner ring, 1), affecting the neighboring regions that became cell-sparse (orange, middle ring, 2), though sparing the lateral-most regions (yellow, outer ring, 3). (**C**) Glutamate concentration within the media associated with samples that were injured increased markedly within 24 h of the initial injury (** *p* < 0.001) and remained elevated for 3 days post-injury (***p* < 0.001); after changing the media (gray dotted line) and incubating for 3 more days, the glutamate concentrations remained elevated in media associated with injured samples relative to the controls (* *p* < 0.005). (**D**) Time-course imaging revealed that live cell populations along the top surface of the scaffolds gradually receded away from the central window (cw) when the samples were injured and untreated, ultimately becoming reduced to patches of cells along the lateral portions of the tissue constructs (red arrows); however, the edge surface remained largely unaffected. Means ± SEMs are provided. All image scale bars represent 500 µm.

**Figure 3 biomolecules-10-01196-f003:**
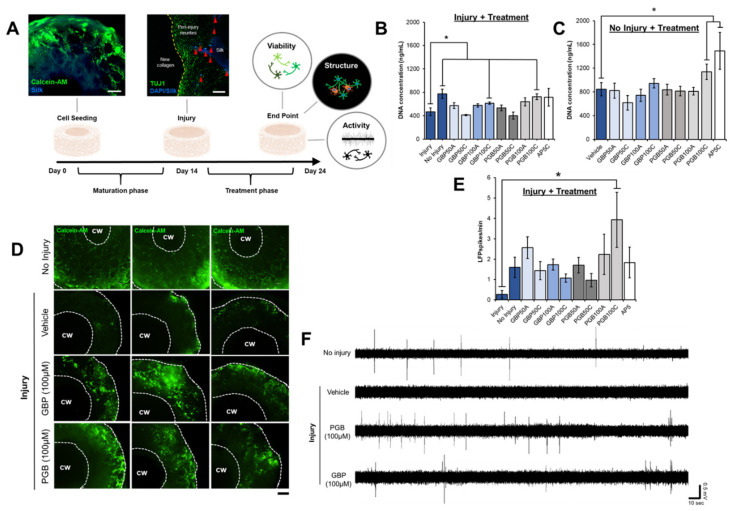
Treatments effects upon injured and non-injured 3D neural tissues. (**A**) Cells were seeded on Day 0 and matured for 14 days, injured, and received treatment for up to 10 days post-laceration after which they were assessed. (**B**) DNA concentration in injured and (**C**) non-injured samples associated with acute and chronic exposures of GBP (50 µM or 100 µM), PGB (50 µM or 100 µM), and AP5 (100 µM) assessed 10 days post-injury are displayed. Acute and chronic treatments are indicated by A and C, respectively. (**D**) Live imaging confirmed the presence of increased live cell populations in injured samples exposed chronically to high concentrations (100 µM) of GBP and PGB relative to those treated with the vehicle. White dotted lines encircle the outer perimeter of the sample as well as the inner perimeter of the central window (cw). The scale bar represents 500 µm. (**E**) Local field potentials (LFPs) recorded from the injured samples (*n* = 9/group) 10 days post-injury revealed a marked reduction in electrical activity (LFP spikes/min) relative to the non-injured controls, a phenomenon which was reversed by a chronic administration of 100 µM PGB (*p <* 0.05). (**F**) Representative traces for the main conditions are provided with scales (20 s, 0.5 mV). Means ± SEMs and the significant differences between groups are indicated (* *p <* 0.05).

**Figure 4 biomolecules-10-01196-f004:**
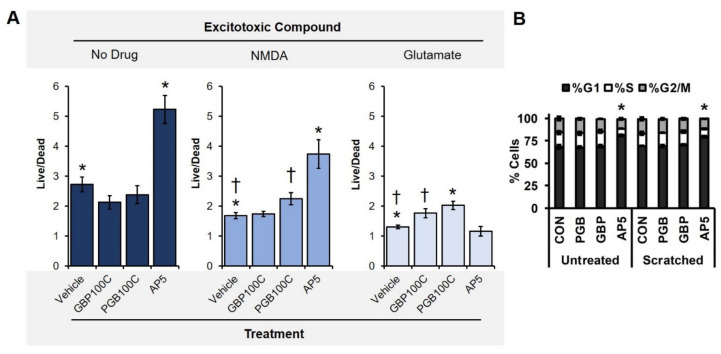
Live–dead imaging of primary cortical neurons exposed to excitotoxic compounds and drug conditions with confirmation of non-proliferation. (**A**, **left**) Chronic exposures to AP5 improved cell viability without the addition of an excitotoxic compound, mirroring effects in 3D reported elsewhere. (**A**, **middle**) When NMDA was added to the culture media, both AP5 and PGB (100C) prevented excitotoxicity. (**A**, **right**) Glutamate-induced toxicity, however, was only prevented by VGCC blockers. Only significant differences relative to the vehicle condition (control, no treatment) are indicated (* *p <* 0.001, † *p <* 0.05). (B) Percentage of primary cortical cells in the respective phases of the cell cycle after 24 h treatment with or without the shown perturbations. VGCCs showed no clear proliferative effect upon cells with or without the simulated injury (scratch); a significant main effect of AP5 on each cell cycle metric compared to all other treatment groups was noted (* *p <* 0.001).

**Figure 5 biomolecules-10-01196-f005:**
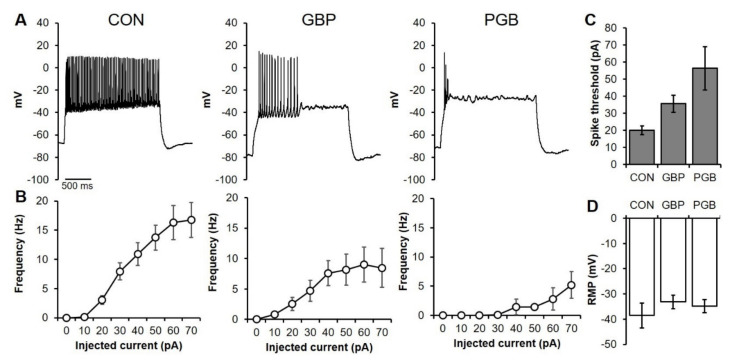
Electrophysiological characterization of primary cortical neurons which were chronically exposed to VGCC blockers. (**A**) Representative traces from intracellular recordings of neurons which were exposed to media only (CON, **left**), or chronic administrations of either 100 µM GBP (**middle**) or 100 µM PGB (**right**). (**B**) Frequency of action potentials upon injection of a current are plotted for each condition. (**C**) Mean spike thresholds and (**D**) mean resting membrane potentials are shown. (CON *n =* 6; GBP, *n =* 9; PGB, *n =* 8).

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
