# Peer review of "A 3D Tissue Model of Traumatic Brain Injury with Excitotoxicity That Is Inhibited by Chronic Exposure to Gabapentinoids"

_biomolecules, 2020, doi:10.3390/biom10081196_

Round 1
Reviewer 1 Report
Biomolecules-848137
Study from Kaplan’s lab established a 3D neural tissue model by growing primary cortical cells in the fabricated silk-based scaffold. To model a laceration-based TBI, they applied surgical biopsy punches to this 3D neural tissue. They characterized their model and further investigate mechanisms of how to protect neurons from TBI-induced excitotoxicity. The manuscript is well written, and the results were acquired with high quality techniques. There are some concerns need to be clarified:
- The authors mentioned that their model is a laceration-based TBI model, have the author compare the pathology of their model with human samples or animal model of laceration-based TBI ? Please provide data/ explanation or discussion.
- The DNA content were decreased in the control group (Fig.2) and could be prevented by AP5 treatment (Fig.3) suggested that this model is quite damage from the processes. How the author compare or verify that their system is greater than the organotypic slice cultures that the author explain in the introduction part? Is this study provide a true non-injured control condition?
- Figure 2D and 3D are very informative, however, it will complete these figures if the author provide a quantitative distance measurement.
- The authors have the list of GFAP antibody in the method section, did the authors observe astrocyte phenotype at post-injury? Please provide result, explain or discuss.
- PGB is very protective and it could prevent the loss of LFPs (Fig.3), however, it reduces firing capability and shifts the curve to the right (Fig.5). Could the author provide insight explanation and discussion for this interaction?
Author Response
Reviewer 1
1.The authors mentioned that their model is a laceration-based TBI model, have the author compare the pathology of their model with human samples or animal model of laceration-based TBI ? Please provide data/ explanation or discussion.
Response: The model presented in the paper does recapitulate TBI pathology presented in the literature; however, the original discussion section did not reference specific studies with which to draw comparisons to our results. The revised manuscript includes several new references in the discussion section which draw comparisons to the presented data.
2.The DNA content were decreased in the control group (Fig.2) and could be prevented by AP5 treatment (Fig.3) suggested that this model is quite damage from the processes. How the author compare or verify that their system is greater than the organotypic slice cultures that the author explain in the introduction part? Is this study provide a true non-injured control condition?
Response: The model system was intrinsically limited by the longevity of primary rat neuron cultures, which explains the DNA decreases exhibited by controls. While this does represent a kind of deterioration, it is notably not comparable to the laceration-based effects as indicated by both the DNA and glutamate results presented in Figure 2. We address this in the discussion section: “We interpret these findings to reflect one limitation associated with the long-term culture of primary cortical cells derived from rodents. After approximately 21-24 days post-seeding, the 3D constructs began to slowly deteriorate, whether injured or not.”. We then propose future studies should use human stem cells as we have demonstrated previously: “Rodent-based neural tissue systems are therefore only suitable for short-term (<21 day) experiments. Human induced pluripotent stem cells (iPSC) and neural stem cells (h thNSC) are more suitable for long-term applications and can be readily integrated within the 3D silk scaffolds (Cairns et al., 2018; Cantley et al., 2018).”. The paper also includes reference to our recent paper demonstrating stability of human-based tissues over 2 years in culture (Rouleau et al., 2020).
3.Figure 2D and 3D are very informative, however, it will complete these figures if the author provide a quantitative distance measurement.
Response: The examples provided in 2D and 3D have poorly-defined “fronts” that would not be conducive to quantification of distance (from some central reference point). The images are provided as qualitative, supplementary visualization of the cell drop-out noted by quantifying DNA within lacerated and non-lacerated scaffolds.
4.The authors have the list of GFAP antibody in the method section, did the authors observe astrocyte phenotype at post-injury? Please provide result, explain or discuss.
Response: The inclusion of GFAP was an error during the writing phase. Mention of GFAP has been removed from the revised manuscript.
- PGB is very protective and it could prevent the loss of LFPs (Fig.3), however, it reduces firing capability and shifts the curve to the right (Fig.5). Could the author provide insight explanation and discussion for this interaction?
Response: These effects were anticipated and are consistent with the known mechanisms underlying the effects of PGB. References and the relevant literature have been added in the revised manuscript. The rightward shift is explained by a necessity to increase injected current to elicit even a modest neural response in the presence of PGB. Again, this is consistent with the known mechanisms that underlie the effects of PGB. The revised manuscript provides these explanations and the relevant literature.
Reviewer 2 Report
In this manuscript, Rouleau and co-authors demonstrated the potential application of a silk-collagen based bioengineered 3D culture system for cortical brain model of traumatic brain injury (TBI) resembles primary and secondary injury. The manuscript displays well developed thought process implemented with appropriate experiments. This reviewer has few suggestions/comments that the authors may consider address and revise the manuscript where applicable.
- In the discussion section, the authors mentioned rodent-based neural tissue systems are only suitable for short-term (<21 day) experiments, while human induced pluripotent stem cells (iPSC) and neural stem cells (HNSC) are more suitable for long-term applications and can be readily integrated within the 3D silk scaffolds. Since these findings were already published from the same group, why did not the authors use hiPSC or HNSC?
- The authors used cortical neuron cells derived from rat embryos (E18). How is this embryonic cell model similar to adult traumatic brain injury (TBI)? Will there be different outcome if the authors use adult neuron cells? Will it be incorrect to assume that embryonic cells will display far better response compared to fetal/adult neuron cells? The authors should discuss details about this.
- Did the authors measure any pro- and anti-inflammatory cytokines in response to the injury in the 3D model? Can the authors shed some light in this perspective?
- Do the authors have any idea/plan if there are any macrophages associated with TBI?
- The authors did not show different types of cell (surface or internal) markers associated with TBI. Why is that?
- The authors failed to discuss limitations of this model. Please discuss this in the discussion section. This will provide valuable insights to the scientific community.
Author Response
Reviewer 2:
1.In the discussion section, the authors mentioned rodent-based neural tissue systems are only suitable for short-term (<21 day) experiments, while human induced pluripotent stem cells (iPSC) and neural stem cells (HNSC) are more suitable for long-term applications and can be readily integrated within the 3D silk scaffolds. Since these findings were already published from the same group, why did not the authors use hiPSC or HNSC?
Response: The current work as well as the referenced works by both Cairns et al., and Cantley et al., were completed in parallel. Rodent cells were selected for the present studies due to the significant troubleshooting associated with assaying several laceration methods simultaneously. Primary cells are available in abundance relative to stem cells. Even though long-term studies would clearly benefit from more robust long-term cultures, the rodent cells were still useful in recapitulating secondary injury mechanisms and were susceptible to known treatments.
2.The authors used cortical neuron cells derived from rat embryos (E18). How is this embryonic cell model similar to adult traumatic brain injury (TBI)? Will there be different outcome if the authors use adult neuron cells? Will it be incorrect to assume that embryonic cells will display far better response compared to fetal/adult neuron cells? The authors should discuss details about this.
Response: The E18 rat cells were used as they were part of a previously established cortical brain model (e.g., Chwalek et al., 2015, Nature Protocols, 10(9)). Adult neurons are not suitable candidates for isolation and cell culture, either in 3D or 2D (petri dish). We have previously used perinatal neurons (P1,P2); however, embryonic cells are much less likely to die immediately after the seeding procedure and based upon our electrophysiological data presented in Figure S2, the cells are capable of eliciting action potentials. Similarly, the maturity of the embryonic cells can be inferred based upon the presence of voltage-gated channels and glutamatergic activity (Figure 3), as well as their 2D morphology (Figure S3), and post-mitotic phenotypes (Figure 4B). It is correct to assume that embryonic cells are more resilient to injury relative to adult cells, however, maturation is evident. Also, Figure S1 displays the 2 mm laceration injury, which was insufficiently severe to elicit the TBI-like phenotype. We therefore opted for a 3 mm, more severe injury. We have included some information relevant to these points in the discussion section of the revised manuscript.
3.Did the authors measure any pro- and anti-inflammatory cytokines in response to the injury in the 3D model? Can the authors shed some light in this perspective?
Response: We have previously measured cytokines from similar tissues (Liaudanskaya et al., 2020, doi: 10.1002/adhm.202000122) under injury conditions. However, the focus of the current study was on the excitotoxic mechanism and its reversal. Therefore, we limited assays to narrow the scope of the study and to preserve sample size.
4.Do the authors have any idea/plan if there are any macrophages associated with TBI?
Response: We are planning to integrate more complex processes in future studies including the role of macrophages; however, the current paper was focused on a specific mechanism underlying secondary injury and its reversal.
5.The authors did not show different types of cell (surface or internal) markers associated with TBI. Why is that?
Response: The main marker that we were interested in was extracellular glutamate release associated with the secondary injury phase of the simulated TBI. Because we were also interested in cultures at the “tissue level” (i.e., macroscopic), we also measured the spatial distribution of cell death over time following injury. While markers of necrosis, apoptosis, and inflammation would have certainly been interesting, we were more focused on excitotoxicity as a downstream effect of a simulated brain injury in vitro and whether it could be reversed or inhibited pharmacologically.
6.The authors failed to discuss limitations of this model. Please discuss this in the discussion section. This will provide valuable insights to the scientific community.
Response: The revised manuscript now includes further details with regards to the limitations of the model system, please see revised discussion section.
Reviewer 3 Report
The article "A 3D tissue model of traumatic brain injury with excitotoxicity that is inhibited by chronic exposure to gabapentinoids" is devoted to solving an important problem in modern medicine, namely, the mechanisms of brain damage in traumatic brain injury. The authors proposed a 3D tissue model of traumatic brain injury (a laceration-based TBI model which using 3D bioengineered neural tissues from cortex) using the scaffold developed by the authors. The proposed model is original and adequate and allowed the authors to study some mechanisms of damage and repair of brain neurons. In particular, the authors studied the effects of gabapentinoids as inhibitors of the voltage-gated calcium channel.
The results obtained are significant, all conclusions supported by received results. Thus, the authors on the proposed model confirmed the most common hypothesis of glutamate excitotoxicity of neuronal death follow TBI.
The article is framed correctly. The results obtained are reliable.The methods used are well known and can be reproduced by other researchers.
In this article, the main mechanisms of neuronal damage have been consistently studied, and it has been also shown that chronic administration of gabapentinoids delays neuronal death.
There are comments on the article:
- It is advisable that in all figures where statistically processed results are given, indicate their significance, and not just cite the text
- The authors convincingly proved that in the model of traumatic brain injury that they used, the main mechanism causing neuronal death is glutamate, the concentration of which in the applied model increases significantly. Glutumate causes overstimulation of glutamate receptors and, as a result, significantly increases [Ca2 +]i, develops [Ca2 +]i-plateau, the main trigger of neuronal death. To protect neurons, the authors used gabapentinoids as inhibitors of the voltage-gated calcium channels, which reduce Ca2 + influx. Therefore, the article would become even more interesting if the authors would receive data on changes in [Ca2 +]i in model of TBI and under the action of gabapentinoids. It's easy to do with a confocal microscope and fluorescent probes.
Author Response
Reviewer 3:
- It is advisable that in all figures where statistically processed results are given, indicate their significance, and not just cite the text
Response: Statistical significance values have been included with figures in the revised manuscript as per the reviewer’s suggestion.
- The authors convincingly proved that in the model of traumatic brain injury that they used, the main mechanism causing neuronal death is glutamate, the concentration of which in the applied model increases significantly. Glutumate causes overstimulation of glutamate receptors and, as a result, significantly increases [Ca2 +]i, develops [Ca2 +]i-plateau, the main trigger of neuronal death. To protect neurons, the authors used gabapentinoids as inhibitors of the voltage-gated calcium channels, which reduce Ca2 + influx. Therefore, the article would become even more interesting if the authors would receive data on changes in [Ca2 +]i in model of TBI and under the action of gabapentinoids. It's easy to do with a confocal microscope and fluorescent probes.
Response: We have only recently validated calcium imaging in our tissues comprised of primary mouse neurons (not rats) using the calcium reporter GCAMP6f (Yu-Ting et al., 2020 in iScience, doi: 10.1016/j.isci.2020.101434). We have not performed similar necessary validations or characterizations with rat cells or in lacerated samples; however, the current paper has used pharmacological methods to infer the role of calcium.
Reviewer 4 Report
The traumatic brain injury (TBI) associated with brain rupture, undoubtedly, insidious. In vitro models of TBI helps to clarify the pathological mechanisms responsible for cell death. The authors proposed a novel bioengineered cortical brain model of TBI (a laceration-based TBI model which using 3D bioengineered neural tissues from cortex) that displays characteristics of primary and secondary injury including an outwardly-radiating cell death phenotype and increased glutamate release with excitotoxic features. The scaffold-hydrogel hybrid structure allows the user to control material properties including porosity and stiffness, generate internal compartments, and deliver significant lacerations while preserving the contour of the simulated tissues. In addition, the new model of TBI has shown the effectiveness of the standard therapy used in the clinic and is therefore quite adequate. The article will be of interest to researchers who study the mechanisms of neurons damage and repair in traumatic brain injury. No questions.
Author Response
Reviewer 4:
No questions/No responses
Reviewer 5 Report
The authors provide a robust and comprehensive, yet concise background on TBI pathophysiology and strongly motivate the need for engineering 3D models with appropriate references and context. The authors show the ability to create a 3D tissue engineered model that replicate some of the hallmarks of TBI in vivo. The authors use a combination of imaging, DNA quantification, and elecotrophysiology techniques to confirm cell death that penetrates from the injury core to penumbra, resulting in increased glutamate release. The authors showed, in response to PGB treatment, they achieved a neuroprotective effect, and confirmed this response was not due to proliferation of cells. The detailed and transparent statistical analysis and reporting of contributions to variance, are appreciated. This paper demonstrates a unique tool for studying brain injury and screening therapies for the TBI field, and should be accepted for publication after addressed the comments listed below.
Comments
Page 8, line 317-218: The authors mention both neurons and glia were present in the model. What was the process for confirming glial cells were present, and what type of glial cells were incorporated? Additionally, do the authors know the ratio of neural to glial cells in their model?
Page 10, line 390-391: the authors note that glutamate concentrations are higher than non-injured controls even after media changes. Did the authors identify whether media changes were removing dead cells?
Methods - immunocytochemistry and live cell imaging, page 6: Can the authors add their method or process for thresholding fluorescent images for both PFA fixed samples and live cell imaging samples? Was an Otsu method (or comparable) used to identify and then subtract background?
Methods - excitotoxicity assay, page 7 - what form of glutamate was used to induce cytotoxicity in primary cortical neurons in 2D culture? Glutamate comes in salt form (MSG) or acid form, both of which can induce additional injury or toxicity when applied to cells (dose-dependent, of course).
Page 10, line 410-411: the authors state "we electrophysiologically characterized the immediate and sustained effects of both drugs.." but only list GBP in the first part of the sentence. For clarity, can the authors edit this sentence to identify the two drugs that are being tested?
Discussion, page 15, line 562-571: the authors highlight the needs of in vitro models, but don't explicitly state the limitations of their model system, other than the decrease of DNA on lines 579-580. Can the authors comment on these limitations, particularly the lack of interstitial flow, and their models compatibility with a flow or perfusion setup?
Author Response
Reviewer 5:
Page 8, line 317-218: The authors mention both neurons and glia were present in the model. What was the process for confirming glial cells were present, and what type of glial cells were incorporated? Additionally, do the authors know the ratio of neural to glial cells in their model?
Response: The inclusion of GFAP was an error during the writing phase. Mention of GFAP has been removed from the revised manuscript. Glia are, however, known to be present in the tissue model system and have been previously measured. The revised manuscript cites specific papers that have already characterized glial cell populations within the model system.
Page 10, line 390-391: the authors note that glutamate concentrations are higher than non-injured controls even after media changes. Did the authors identify whether media changes were removing dead cells?
Response: Dead cells in removed media were not quantified. However, the removal of dead cells from the lacerated tissues would be predicted to decrease the concentration of calcium in the media. It is therefore likely that dead cells remained confined to the hydrogel substrate or were locally dissociated.
Methods - immunocytochemistry and live cell imaging, page 6: Can the authors add their method or process for thresholding fluorescent images for both PFA fixed samples and live cell imaging samples? Was an Otsu method (or comparable) used to identify and then subtract background?
Response: Further details concerning thresholding have been added to the methods section. We did not use automated methods, however, they were applied consistently across samples and conditions with appropriate counterbalancing.
Methods - excitotoxicity assay, page 7 - what form of glutamate was used to induce cytotoxicity in primary cortical neurons in 2D culture? Glutamate comes in salt form (MSG) or acid form, both of which can induce additional injury or toxicity when applied to cells (dose-dependent, of course).
Response: Glutamate was prepared by solubilizing powdered L-glutamic acid (≥99% (HPLC)) dissolved in water with constant stirring and applied heat. These details were added to the methods section in the revised manuscript.
Page 10, line 410-411: the authors state "we electrophysiologically characterized the immediate and sustained effects of both drugs.." but only list GBP in the first part of the sentence. For clarity, can the authors edit this sentence to identify the two drugs that are being tested?
Response: Yes, this is sensible. The revised manuscript now includes mention of both drugs.
Discussion, page 15, line 562-571: the authors highlight the needs of in vitro models, but don't explicitly state the limitations of their model system, other than the decrease of DNA on lines 579-580. Can the authors comment on these limitations, particularly the lack of interstitial flow, and their models compatibility with a flow or perfusion setup?
Response: The revised manuscript now includes further details with regards to the limitations of the model system.